# Residual Matrix Transformers: Scaling the Size of the Residual Stream

**Brian Mak** [1]   **Jeffrey Flanigan** [1]

## Abstract

The residual stream acts as a memory bus where transformer layers both store and access features (Elhage et al., 2021). We consider changing the mechanism for retrieving and storing information in the residual stream, and replace the residual stream of the transformer with an outer product memory matrix (Kohonen, 1972, Anderson, 1972). We call this model the Residual Matrix Transformer (RMT). We find that the RMT enjoys a number of attractive properties: 1) the size of the residual stream can be scaled independently of compute and model size, improving performance, 2) the RMT can achieve the same loss as the transformer with 58% fewer FLOPS, 25% fewer parameters, and 41% fewer training tokens tokens, and 3) the RMT outperforms the transformer on downstream evaluations. We theoretically analyze the transformer and the RMT, and show that the RMT allows for more efficient scaling of the residual stream, as well as improved variance propagation properties. Code for this project can be found at https://github.com/bmac3/residual-matrix-transformer.

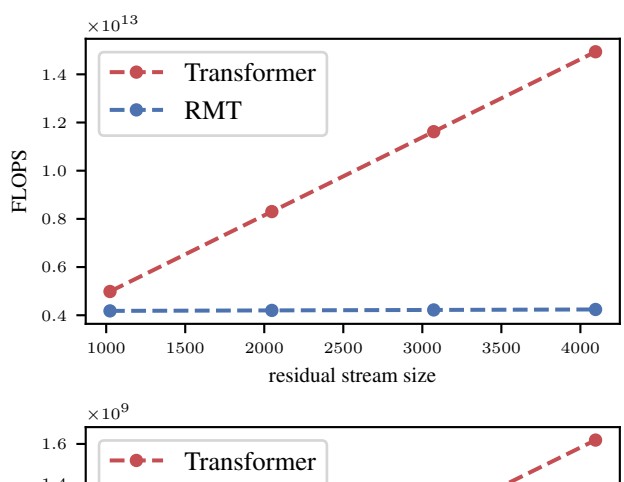

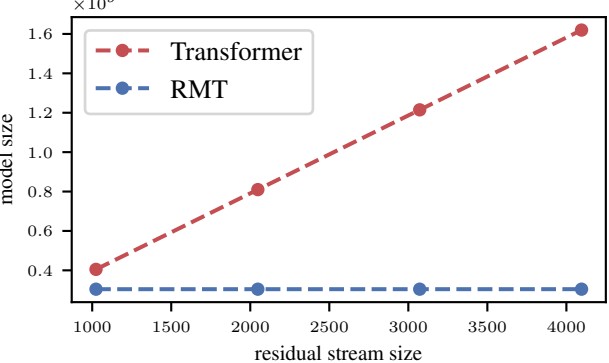

*Figure 1.* Model size and per-example FLOPS versus residual stream size for the transformer and RMT. See §3.1 for details.

## 1. Introduction

Kaplan et al. (2020) presented a new way forward for the field of NLP and AI, showing that simply scaling model size, data size, and computational budget is sufficient to advance model capabilities at an unprecedented rate. Their work lead to the the creation of many of the frontier models today (Achiam et al., 2023, Hoffmann et al., 2024, Dubey et al., 2024). Meanwhile, data, compute, and model sizes continue to grow exponentially each year (AI, 2024). In fact, we are quickly scaling towards the limits on how much data and energy is available (Muennighoff et al., 2023, Zachary Zim-

merman & Gramlich, 2023). This presents a need for more data and compute efficient models.

When it comes to large scale LLM training, few architectural modifications rival the impact that Mixture of Expert Models (Fedus et al., 2022) have had. Their work inspired many later cutting-edge models (Du et al., 2022, Jiang et al., 2024). Their Switch Transformer model had the ability to scale along the sparse parameter axis. This meant that model size could be scaled *without affecting per-example compute.* With this, they were able to scale LLM training in ways that had previously been unexplored. In this new search space, they found a training configuration that was 7 times more efficient than standard transformer training.

In this paper, we follow a similar formula and present a new transformer-type model, the Residual Matrix Transformer

[1]Department of Computer Science, University of California Santa Cruz, Santa Cruz, United States. Correspondence to: Brian Mak <bmak2@ucsc.edu>.

*Proceedings of the 42^{nd} International Conference on Machine Learning*, Vancouver, Canada. PMLR 267, 2025. Copyright 2025 by the author(s).

(RMT), that can scale the residual stream size *while keeping model size and per-example compute fixed*. The **residual stream** is defined as the sum of all the outputs of the previous layers and acts as a communication channel between layers (Elhage et al., 2021). The size (i.e. dimension) of the residual stream determines how many features it can store, so scaling its size can be thought of as scaling its bandwidth. In our experiments, we find that scaling the size of the residual stream while maintaining the constant model size and per-example compute improves both data and compute efficiency (§4.4).

Scaling the residual stream in a standard transformer resizes all the parameter matrices, which linearly influences the parameter and FLOP counts (see Fig. 1). To address this, we replace the residual stream representation with an outer product memory matrix (Kohonen, 1972, Anderson, 1972), and find the relationship between the size of the residual stream and parameter and FLOP count becomes nearly constant for typical transformer dimensions. Additionally, we find that the our proposed model is more efficient in terms of data, FLOPS, and parameters than the standard transformer.

We summarize our contributions as follows:

- We introduce a new transformer-type model, the Residual Matrix Transformer (RMT), that replaces the residual stream with an outer product memory matrix (§2).

- We experimentally find that the RMT is more efficient in terms of data, FLOPS, and parameters than the transformer. The RMT trains with fewer resources and achieves better performance than the transformer and variants (§4.2, §4.3, and §4.5).

- We present theory that shows the RMT exhibits efficient scaling of the residual stream and has improved variance propagation properties (§3).

- We experimentally explore the effects of scaling the residual stream size while keeping model size and per-example compute fixed with the RMT. We find that a larger residual stream improves performance (§4.4).

## 2. Model Architecture

In this section we introduce the Residual Matrix Transformer, a transformer variant whose residual stream is replaced with an outer product memory mechanism. We first review the transformer architecture (§2.1), and then present the RMT formulation (§2.2).

### 2.1. Transformer

**Transformer Overview**    We begin by giving the equations that describe the transformer architecture (Vaswani et al., 2017). In our formulation, the input to the transformer is a matrix $\boldsymbol{S} \in \mathbb{R}^{V \times N}$ whose columns are the one-hot-vectors of some tokenized string. The first layer encodes the input.

$$\boldsymbol{X}^{(0)} = \mathrm{E}(\boldsymbol{S}) + \mathrm{P}(\boldsymbol{N}) \tag{1}$$

Here, $\boldsymbol{N} \in \mathbb{R}^{N \times N}$ are the one-hot-vectors representing each token's position. The output of this layer is an embedding matrix $\boldsymbol{X}^{(0)} \in \mathbb{R}^{D \times N}$, and it contains the first instance of each token's residual stream. The next $2L$ layers are defined recursively and describe the successive application of attention and feed-forward layers to the residual stream.

$$\boldsymbol{X}^{(l)} = \mathrm{MHA}(\mathrm{LN}(\boldsymbol{X}^{(l-1)})) + \boldsymbol{X}^{(l-1)} \tag{2}$$

$$\boldsymbol{X}^{(l+1)} = \mathrm{FF}(\mathrm{LN}(\boldsymbol{X}^{(l)})) + \boldsymbol{X}^{(l)} \tag{3}$$

where LN is LayerNorm (Xu et al., 2019). The final layer reads out the logits of the next predicted token from the residual stream. The transformer T is thus represented as:

$$\mathrm{T}(\boldsymbol{S}) = \mathrm{U}(\mathrm{LN}(\boldsymbol{X}^{(2L)})) \tag{4}$$

**Embedding Layer**    The embedding layer encodes the one-hot-vector matrices using the learned embeddings $\boldsymbol{W}_E \in \mathbb{R}^{D \times V}$ and $\boldsymbol{W}_{PE} \in \mathbb{R}^{D \times N}$.

$$\mathrm{E}(\boldsymbol{S}) = \boldsymbol{W}_E \boldsymbol{S} \tag{5}$$

$$\mathrm{P}(\boldsymbol{N}) = \boldsymbol{W}_{PE} \boldsymbol{N} \tag{6}$$

**Attention Layer**    For our attention layer definition, we group the computation of each attention head separately.

$$\mathrm{MHA}(\boldsymbol{X}) = \sum_{h=1}^{H} \boldsymbol{W}_O^{(h)} \mathrm{SHA}(\boldsymbol{Q}^{(h)}, \boldsymbol{K}^{(h)}, \boldsymbol{V}^{(h)}) \tag{7}$$

$$\mathrm{SHA}(\boldsymbol{Q}, \boldsymbol{K}, \boldsymbol{V}) = \mathrm{Softmax}(\frac{\boldsymbol{Q}^T \boldsymbol{K}}{\sqrt{D_h}})\boldsymbol{V} \tag{8}$$

$$\boldsymbol{Q}^{(h)} = \boldsymbol{W}_Q^{(h)} \boldsymbol{X} \quad \boldsymbol{K}^{(h)} = \boldsymbol{W}_K^{(h)} \boldsymbol{X} \quad \boldsymbol{V}^{(h)} = \boldsymbol{W}_V^{(h)} \boldsymbol{X} \tag{9}$$

$\boldsymbol{W}_Q^{(h)}, \boldsymbol{W}_K^{(h)}, \boldsymbol{W}_V^{(h)} \in \mathbb{R}^{D_h \times D}$ and $\boldsymbol{W}_O^{(h)} \in \mathbb{R}^{D \times D_h}$.

**FeedForward Layer**    We define our feed-forward network in the standard way, using a Gelu activation. $\boldsymbol{W}_1 \in \mathbb{R}^{D_{FF} \times D}$ and $\boldsymbol{W}_2 \in \mathbb{R}^{D \times D_{FF}}$.

$$\mathrm{FF}(\boldsymbol{X}) = \boldsymbol{W}_2 \mathrm{Gelu}(\boldsymbol{W}_1 \boldsymbol{X}) \tag{10}$$

**Unembedding Layer**    The unembedding layer multiplies the final instance of the residual stream with unembedding weights $\boldsymbol{W}_U \in \mathbb{R}^{V \times D}$.

$$\mathrm{U}(\boldsymbol{X}) = \boldsymbol{W}_U \boldsymbol{X} \tag{11}$$

## 2.2. Residual Matrix Transformer

Here we give relevant technical background on outer product memories and show how this mechanism can be applied to the transformer.

**Outer Product Memories**  We propose to replace the residual stream with an outer product memory store. An outer product memory store is created by summing the outer products of a set of key-value vector pairs (Kohonen, 1972, Anderson, 1972, Gmitro et al., 1989). For some set of key vectors $\{q^{(p)} \in \mathbb{R}^{D_k}\}_{p=1}^N$ and data vectors $\{x^{(p)} \in \mathbb{R}^{D_v}\}_{p=1}^N$, an outer product store $M \in \mathbb{R}^{D_k \times D_v}$ can be constructed using the following equation:

$$M = \text{Norm}(\sum_{p=1}^N q^{(p)} \otimes x^{(p)}) \qquad (12)$$

Here $u \otimes v = uv^T$. Norm is a normalization function, which in the case of RMT is LayerNorm. One can retrieve a particular data vector $x^{(r)}$ using its corresponding key vector $q^{(r)}$:

$$x^{(r)} \approx q^{(r)} \cdot_1 M \qquad (13)$$

Here $\cdot_1$ denotes tensor contraction over the first dimension of $q^{(r)}$ and $M$.

**RMT Overview**  The RMT is similar to the Transformer, with the residual stream vectors replaced with outer product memory stores.[1] We refer to each token's memory store as its residual matrix. Whereas in the standard transformer linear transformations are used to store and retrieve features from the residual stream, our model uses key vectors to store and retrieve data vectors from the residual matrix.

The RMT model is identical to Eqs. 1–4, except now the batched residual stream instances $\mathbf{X}^{(l)} \in \mathbb{R}^{D_k \times D_v \times N}$ are tensors instead of matrices. This reflects the fact that in our model, for every token position we have residual matrices sized $D_k \times D_v$ instead of residual stream vectors sized $D$.

$$\mathbf{X}^{(0)} = \text{E}(S) + \text{P}(N) \qquad (14)$$

$$\mathbf{X}^{(l)} = \text{MHA}(\text{LN}(\mathbf{X}^{(l-1)})) + \mathbf{X}^{(l-1)} \qquad (15)$$

$$\mathbf{X}^{(l+1)} = \text{FF}(\text{LN}(\mathbf{X}^{(l)})) + \mathbf{X}^{(l)} \qquad (16)$$

$$\text{T}(S) = \text{U}(\text{LN}(\mathbf{X}^{(2L)})) \qquad (17)$$

---

[1]More precisely, the residual stream vectors are replaced with unnormalized outer product memory stores and the pre-LayerNorm operations (Xiong et al., 2020) in Eqs. 15–17 provide the missing normalization from Eq. 12.

**Embedding Layer**  Eqs. 18 and 19 show how the initial residual matrices are formed.

$$\text{E}(S) = \sum_{h=1}^R w_E^{(h)} \otimes W_E^{(h)} S \qquad (18)$$

$$\text{P}(N) = \sum_{h=1}^R w_{PE}^{(h)} \otimes W_{PE}^{(h)} N \qquad (19)$$

Here, $W_E^{(h)} \in \mathbb{R}^{D_v \times V}$ and $w_E^{(h)} \in \mathbb{R}^{D_k}$ and $R$ is a hyperparameter. Notice the form of Eq. 18 and 19 mirror that of Eqs. 12. The result of the embedding layer is a tensor of shape $D_k \times D_v \times N$ that contains an outer product memory matrix for every token.

**Attention Layer**  The attention equations for the RMT strongly resemble Eqs. 7–9 with a few key differences.

$$\text{MHA}(\mathbf{X}) = \sum_{h=1}^R w_O^{(h)} \otimes \text{SHA}(Q^{(h)}, K^{(h)}, V^{(h)}) \quad (20)$$

$$\text{SHA}(Q, K, V) = \text{Softmax}(\frac{Q^T K}{\sqrt{D_v}})V \qquad (21)$$

$$Q^{(h)} = r_Q^{(h)} \cdot_1 \mathbf{X} \quad K^{(h)} = r_K^{(h)} \cdot_1 \mathbf{X} \quad V^{(h)} = r_V^{(h)} \cdot_1 \mathbf{X} \qquad (22)$$

In the RMT, features are retrieved from the residual matrix using key vectors. Comparing equations Eqs. 22 to Eqs. 9, we see that the matrices $W_Q^{(h)}$, $W_K^{(h)}$, and $W_V^{(h)}$ are replaced by the key vectors $r_Q^{(h)}, r_K^{(h)}, r_V^{(h)} \in \mathbb{R}^{D_k}$. Additionally, the operations have changed from matrix multiplies to tensor contractions over the first tensor dimension, which is exactly the retrieval operation from Eq. 13.

Eqs. 22 produce the attention inputs $Q^{(h)}, K^{(h)}, V^{(h)} \in \mathbb{R}^{D_v \times N}$. These matrices have the exact same dimensions as they did in the standard transformer,[2] so the SHA operation remains the same.

In Eq. 20 we see that each attention head output acts as a data vector that is associated to its corresponding key vector $w_O^{(h)} \in \mathbb{R}^{D_k}$ before being added to the residual matrix.

**FeedForward Layer**  To understand how the RMT feedforward layer is computed, first notice that Eq. 24 is identical to Eq. 10.

$$\text{FF}(\mathbf{X}) = \sum_{h=1}^R w_{FF}^{(h)} \otimes \text{unvec}_1(\widetilde{\text{FF}}(X_{FF}))_{h,:,:} \qquad (23)$$

$$\widetilde{\text{FF}}(X) = W_2 \text{Gelu}(W_1 X) \qquad (24)$$

---

[2]RMT's $D_v$ equals transformer's $D_h$ for our experiments

$$X_{FF} = \underset{1 \le h < R}{\text{Concat}} \left( r_{FF}^{(h)} \cdot_1 \mathbf{X} \right) \qquad (25)$$

That is, the core operation of the layers is the same. The only differences are Eqs. 23 and 25, which are adapters between the residual matrix and the core feed-forward operation. The idea behind Eq. 25 is that for every token position, we retrieve $R$ data vectors $r_{FF}^{(h)} \cdot_1 \mathbf{X} \in \mathbb{R}^{D_v \times N}$ from the residual matrix. These data vectors are concatenated along their first dimensions resulting in the feed-forward input $X_{FF} \in \mathbb{R}^{RD_v \times N}$.[3] In Eq. 24 the standard feed-forward operation is applied. Then in Eq. 23, we see that $\text{unvec}_1$ is applied to $\widetilde{\text{FF}}(X_{FF}) \in \mathbb{R}^{RD_v \times N}$. As a result, the first dimension of $\widetilde{\text{FF}}(X_{FF})$ will be reshaped from $RD_v$ to $R \times D_v$. Intuitively, for every token, $\text{unvec}_1$ splits the output of the feed-forward operation into $R$ data vectors. Then, each of these data vectors is associated with a key vector $w_{FF}^{(h)} \in \mathbb{R}^{D_k}$ and added to the residual stream.

Unlike in the attention formulation, matrices $W_1$ and $W_2$ of the feed-forward layer were not replaced with key vectors. Instead, we added key vector adapters between the linear transformations and the residual matrix. The reason we decided not to replace these matrices is that there is strong evidence that the feedforward weights store factual information in the transformer (Geva et al., 2021, Meng et al., 2022), and therefore perform a function beyond simply reading-in and writing-out features.

**Unembedding Layer**  In Eq. 27, the unembedding layer retrieves $R$ data vectors $X_U^{(h)} \in \mathbb{R}^{D_v \times N}$ from the residual matrix using the key vectors $r_U^{(h)} \in \mathbb{R}^{d_k}$. These data vectors are then multiplied by the unembedding weights $W_U^{(h)} \in \mathbb{R}^{V \times D_v}$ to get logit predictions for the next token.

$$U(\mathbf{X}) = \sum_{h=1}^{R} W_U^{(h)} X_U^{(h)} \qquad (26)$$

$$X_U^{(h)} = r_U^{(h)} \cdot_1 \mathbf{X} \qquad (27)$$

# 3. Theoretical Properties

In §3.1 we show that the RMT allows for more efficient expansion of the residual stream compared to the transformer. Then in §3.2 we show that the elements we modified in the RMT have superior gradient and activation propagation properties than their transformer counterparts.

## 3.1. Resource Scaling

Here we verify that we can scale the size of the residual stream more efficiently with the RMT than the transformer in terms of model size and FLOPS consumed. Fig. 1 shows

how the model size and per-example compute scales with the residual stream size for both the transformer and the RMT. We follow Hoffmann et al. (2024)'s formulas for computing model size and per-example compute in the transformer, and derive the corresponding formulas for the RMT (Appendix §A, Table 7). We fix all model dimensions to those of GPT2-medium[4] (Radford et al., 2019) and vary $D$ for the transformer and $D_k$ for the RMT (where $D_v$ is fixed at 64). We see that the model size and per-example compute for the transformer scales proportionally to the residual stream size. In the RMT, however, there is practically no parameter or compute cost. In fact, if we increase the residual stream size of the transformer by $100\%$, we will also see a $100\%$ increase in parameters and a $\sim 94\%$ increase in FLOPS. For the RMT, if we increase the residual stream size by $100\%$ we will see a $< 1\%$ increase in both model size and FLOP count. We note that, as in the regular transformer, scaling the residual stream of the RMT will impact memory usage during training since residual stream activations need to be saved for gradient checkpointing.

## 3.2. Moment Propagation Analysis

Here we verify that our proposed architectural modifications have superior moment propagation properties to their transformer counterparts. In their landmark work, Glorot & Bengio (2010) showed that proper propagation of mean and variance through linear transformations at initialization is critical for deep learning. Here, we perform a similar analysis to Glorot & Bengio (2010) and Kedia et al. (2024) to find how the mean and variance propagates through the RMT's storage and retrieval operations. We compare their moment propagation properties to the operations they replaced.

In Table 1 we present the closed-form expression for mean and variance propagation through the RMT's storage and retrieval operations at initialization. We find that, as with linear transformations, if weights are initialized independently with zero mean then the propagated mean will be zero. In this respect, the mean propagation properties of these components are the same as the transformer's.

To understand the variance propagation properties, in Table 2 we plug in the model dimensions for GPT2-medium into the closed-form variance expressions in Table 1. Table 2 shows the results of this analysis. We follow Glorot & Bengio (2010) and consider $\frac{\sigma_{x_{out}}^2}{\sigma_{x_{in}}^2} = 1$ and $\frac{\sigma_{g_{in}}^2}{\sigma_{g_{out}}^2} = 1$ to be optimal since it prevents exploding or vanishing gradients or activations. We see that in all cases except for the attention layer's storage operation, the RMT's operations show favorable variance propagation.

---

[3]RMT's $RD_v$ equals transformer's $D$ for our experiments.

[4]We choose GPT2-medium for this example but the general scaling trends will hold true for any model shapes.

*Table 1.* Closed-form expressions for the propagation of mean and variance for storage and retrieval for the RMT. The expressions for the transformer are taken from (Kedia et al., 2024). The derivation the RMT equations can be found in Appendix §B.

| COMPONENT | MODEL | OPERATION | $\mu_{x_{out}}$ | $\sigma^2_{x_{out}}$ | $\mu_{g_{in}}$ | $\sigma^2_{g_{in}}$ |
|---|---|---|---|---|---|---|
| STORAGE | RMT | $\boldsymbol{X}_{out} = \sum_{h=1}^{R} \boldsymbol{w}^{(h)} \otimes \boldsymbol{x}_{in}$ | 0 | $R\sigma^2_w(\sigma^2_{x_{in}} + \mu^2_{x_{in}})$ | 0 | $d_k\sigma^2_w(\sigma^2_{G_{out}} + \mu^2_{G_{out}})$ |
|  | TRANSFORMER | $\boldsymbol{x}_{out} = \boldsymbol{W}\boldsymbol{x}_{in}$ | 0 | $d_{in}\sigma^2_w(\sigma^2_{x_{in}} + \mu^2_{x_{in}})$ | 0 | $d_{out}\sigma^2_w(\sigma^2_{g_{out}} + \mu^2_{g_{out}})$ |
| RETRIEVAL | RMT | $\boldsymbol{x}_{out} = \boldsymbol{w} \cdot_1 \boldsymbol{X}_{in}$ | 0 | $d_k\sigma^2_w(\sigma^2_{X_{in}} + \mu^2_{X_{in}})$ | 0 | $R\sigma^2_w(\sigma^2_{g_{out}} + \mu^2_{g_{out}})$ |
|  | TRANSFORMER | $\boldsymbol{x}_{out} = \boldsymbol{W}\boldsymbol{x}_{in}$ | 0 | $d_{in}\sigma^2_w(\sigma^2_{x_{in}} + \mu^2_{x_{in}})$ | 0 | $d_{out}\sigma^2_w(\sigma^2_{g_{out}} + \mu^2_{g_{out}})$ |

*Table 2.* Theoretical calculation of propagation of variance through storage and retrieval operations on forward and backward passes. We assume Xavier Initialization and GPT2-medium model shapes. Boldface indicates where one model is better than the other.

| LAYER | OPERATION | MODEL | $\frac{\sigma^2_{x_{out}}}{\sigma^2_{x_{in}}}$ | $\frac{\sigma^2_{g_{in}}}{\sigma^2_{g_{out}}}$ |
|---|---|---|---|---|
| ATTN | STORAGE | RMT | 0.4 | 1.6 |
|  |  | TRANSFORMER | **1** | **1** |
|  | RETRIEVAL | RMT | **1.14** | **0.86** |
|  |  | TRANSFORMER | 0.5 | 1.5 |
| FF | STORAGE | RMT | **1** | **1** |
|  |  | TRANSFORMER | 1.6 | 0.4 |
|  | RETRIEVAL | RMT | **1** | **1** |
|  |  | TRANSFORMER | 0.4 | 1.6 |

# 4. Experiments

In §4.2 and §4.3 we compare the RMT to the transformer and other relevant transformer variants and show that the RMT offers more efficient pretraining in terms of parameter, data, and compute efficiency. Then in §4.4 we show that the new scaling axis we introduce (scaling residual stream size) is an effective way to scale LLM training. In §4.5 we evaluate the downstream performance of the RMT and show that it outperforms a transformer that is 33% larger.

## 4.1. Pretraining Details

All models were trained on the OpenWebText dataset (Gokaslan et al., 2019) in the infinite data regime. Full details for all experiments can be found in Appendix §C. To overcome the "Parameter Lottery" (Dey et al., 2024), we performed hyperparameter tuning on both the RMT and transformer using $\mu$Param Transfer (Yang et al., 2024). We use the hyperparameters found for all pretraining experiments except our comparison against transformer variants experiments in §4.3, since it was too computationally demanding to run hyperparameter tuning for all variants. For the experiments found in §4.3, we used standard transformer hyperparameters for all models.

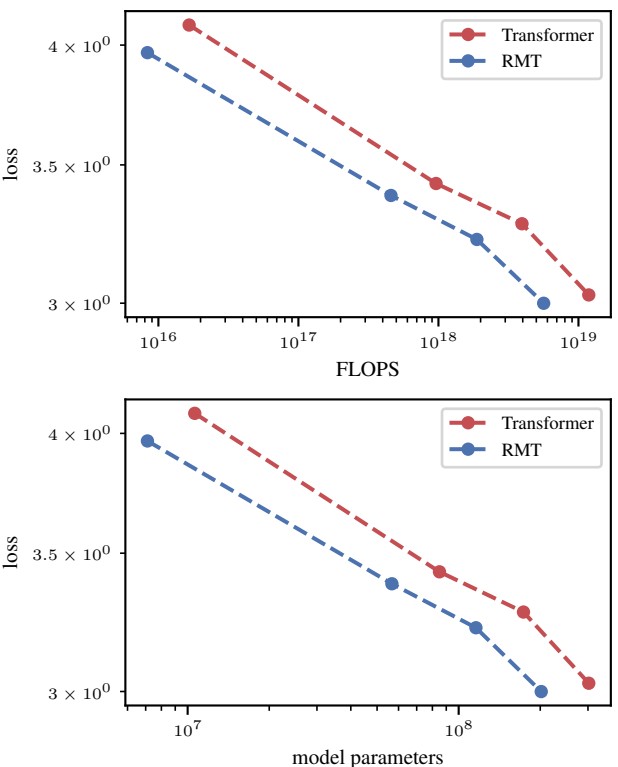

*Figure 2.* Scaling law curves of RMT vs transformer. Model sizes vary from 46M to 405M.

## 4.2. RMT vs Transformer

In this section we experimentally show that the RMT exhibits better parameter, data, FLOP, and inference time memory efficiency than the transformer. We verify these trends by examining the scaling properties of both architectures and analyzing the training curves of the largest models.

We train a series of four models for both architectures. The size of the models vary from 46M parameters to 405M parameters. The model dimensions were chosen such that for each model in the transformer series, there is a corresponding model in the RMT series whose model dimensions mirror it in all aspects except the size of the residual stream.

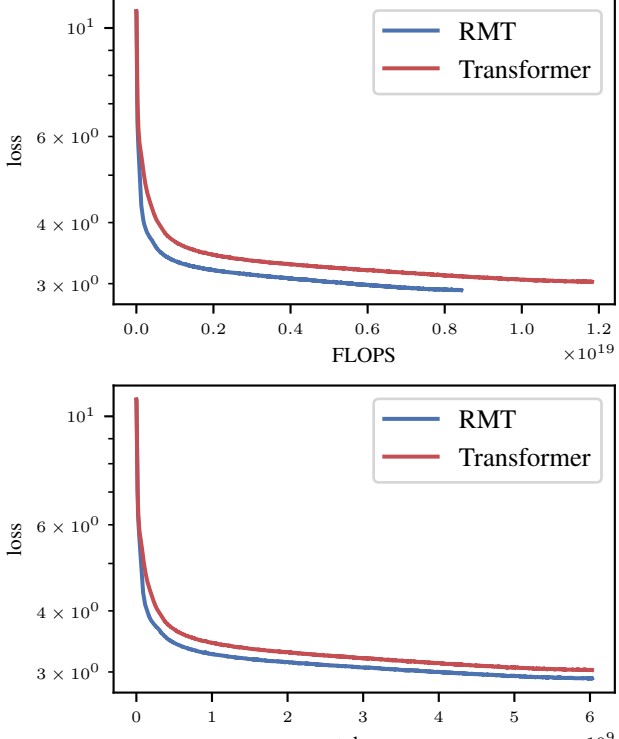

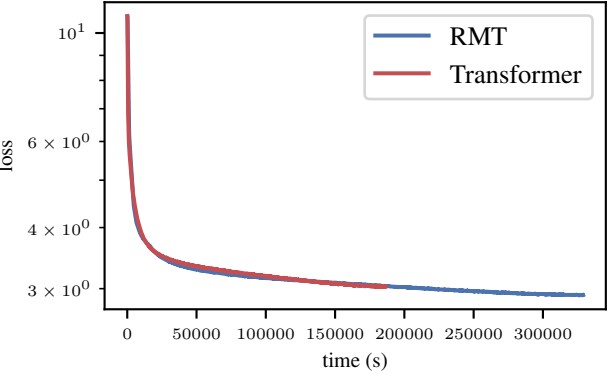

*Figure 4.* Train loss curves for the transformer and RMT on a time basis.

*Table 4.* Training statistics after training on 6B tokens.

| RESOURCE | TRANSFORMER | RMT | % DIFF |
|---|---|---|---|
| FLOPS | $1.18 \times 10^{19}$ | $8.45 \times 10^{18}$ | -28% |
| TIME | $1.86 \times 10^5$ | $3.30 \times 10^5$ | +43% |
| TRAIN LOSS | 3.03 | 2.91 | |
| DEV LOSS | 3.02 | 2.91 | |

*Figure 3.* Train loss curves for the transformer and RMT on a per-token and per-flop basis.

*Table 3.* Resources used to reach train loss 3.03 (min loss achieved by transformer).

| RESOURCE | TRANSFORMER | RMT | % DIFF |
|---|---|---|---|
| PARAMETERS | 405M | 305M | -25% |
| FLOPS | $1.18 \times 10^{19}$ | $4.97 \times 10^{18}$ | -58% |
| TOKENS | $6.04 \times 10^9$ | $3.55 \times 10^9$ | -41% |
| TIME | $1.86 \times 10^5$ | $1.94 \times 10^5$ | +4% |

The residual stream size of the RMT models is set to be 2.5 – 4 times larger than their transformer counterparts. Each model is trained for its Chinchilla optimal number of training tokens (20×number of non-embedding parameters). Fig. 2 shows the results of these experiments. Due to resource constraints, we were unable to explore how these trends continue at larger model sizes.

Fig. 3 and Fig. 4 compare the training runs of the largest models in both series. Both models mirror GPT2-medium's dimensions, except that the residual stream size of the RMT model is expanded by a factor of 4. For a direct comparison, we extended the number of tokens the RMT model was trained on to match the transformer baseline. The results in Fig. 2 still show the RMT's training statistics stopped at its respective Chinchilla optimal train tokens. Table 3

reports the resources consumed to reach the minimum loss achieved by the transformer baseline while Table 4 shows the resources consumed to train on the full 6B tokens.

**Parameter Efficiency** From the bottom plot of Fig. 2 we see that the RMT exhibits more efficient parameter scaling than the transformer. Table 3 and Table 4 quantify this result for our largest runs. We see that the RMT is able to achieve a lower loss than a transformer model that is 33% larger.

The reason why the RMT uses fewer parameters than the transformer is that the $\boldsymbol{W}_Q^{(h)}$, $\boldsymbol{W}_K^{(h)}$, $\boldsymbol{W}_V^{(h)}$, and $\boldsymbol{W}_O^{(h)}$ weight matrices are replaced by vectors in the RMT models. As a result, the RMT models have 33% fewer non-embedding parameters than their transformer counterparts. Furthermore, because of its resource scaling properties from §3.1, we can make the RMT models have a much larger residual stream than their transformer counterparts for practically no parameter penalty.

**Data Efficiency** Fig. 3 shows that the RMT exhibits much faster convergence on a token basis than the transformer. Table 3 quantifies this improvement, showing that the RMT is 41% more token efficient than the transformer. This type of data efficiency is particularly important in the current climate of LLM training, where we are quickly scaling toward the limit of text available to train models on (Villalobos et al., 2024, Muennighoff et al., 2023).

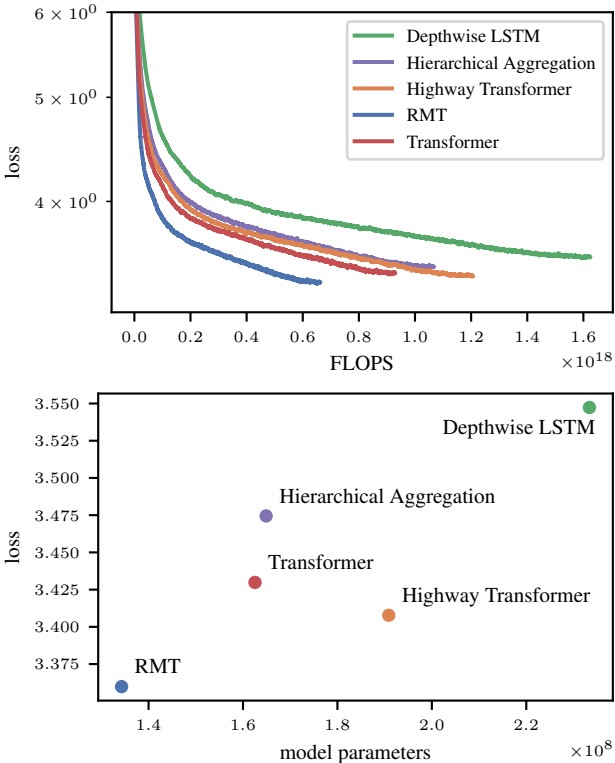

*Figure 5.* RMT versus other transformer variants that modify the residual stream (Vaswani et al., 2017, Dou et al., 2018, Chai et al., 2020, Xu et al., 2024).

**FLOP Efficiency**   From the top plot of Fig. 2 we see that the RMT architecture exhibits more efficient FLOP scaling than the transformer. Fig. 3 corroborates this by showing that on our largest runs the RMT converged faster than the transformer on a per-FLOP basis. Table 3 shows that the RMT is 58% more FLOP efficient when training to the same loss, and Table 3 shows that the RMT is 28% more FLOP efficient when training to the same number of tokens.

We assert that this improved FLOP efficiency presents huge opportunities for energy savings in LLM training. Energy efficiency is another important factor to consider, because like data consumption, energy consumption is also scaling at an untenable rate (Strubell et al., 2019, Luccioni et al., 2023, Wu et al., 2022).

**Memory Usage**   We find that during training, the memory usage of both models is about equal. At inference time, however, the RMT is more memory efficient.

We deduced that the train time memory usage of both models is about equal by performing a batch size search on our largest models. We found that, in both cases, the maximum

batch size found was 256.[5] From this we infer that the extra space needed to store the gradient checkpoints for the expanded residual stream is offset by reduced model size.

At inference time, since there is no need to keep residual stream instances for gradient checkpointing, the memory savings from the reduced model size outweigh the expanded residual stream.[6] This presents opportunities for more powerful models to be run with smaller devices.

**Runtime**   Fig. 4 shows that on a per-time basis, our model is about even with the transformer. Table 3 shows that the RMT is 4% slower than the transformer even though it is more FLOP and data efficient. This discrepancy is due to the fact that the RMT took 7.17s per train step, while the transformer only took 4.06s. We expect that a hardware-aware implementation of our model would significantly improve its runtime, but we consider this is out of the scope of this work. Currently, we find that runtime is the biggest limitation of our model. See Appendix §E for further discussion.

### 4.3. RMT vs Transformer Variants

Here we compare the RMT to transformer variants that can be viewed as non-orthogonal to our work.[7] We find that this class of architectures includes transformer variants that make an architectural modification to the residual stream (Dou et al., 2018, Chai et al., 2020, Xu et al., 2024). We show that our architecture offers more efficient pretraining than all prior variants (Fig. 5).

We train a collection of transformer variants whose model dimensions mirror GPT2-small where applicable. Fig. 5 shows the training curves and final train loss plotted against model size. From the results we see that the RMT is the most compute and parameter efficient variant of this collection, achieving lower loss while using fewer FLOPS and parameters. We note that our approach is distinguished from prior residual stream modifications in that all prior architectures use more parameters and per-example compute than the transformer, while ours uses fewer. Thus, from an efficiency standpoint, the improvements made by the prior variants are outweighed by their cost in our replications.

### 4.4. Effect of Scaling the Residual Stream

In §1 we proposed a new axis to scale LLM training along. In this section we show that this is an effective axis for scaling. In particular, we show that scaling a model's residual stream while keeping model size, dataset size, and computa-

---

[5]The max batch size search was constrained to powers of 2.

[6]Only a single residual matrix needs to be instantiated per sequence for autoregressive decoding. This cost is negligible compared to the size of the replaced attention parameters.

[7]Residual stream modifications can be combined with ours, and so are orthogonal to our work. See related work §5.

tional budget fixed increases model performance. We note that this type of experiment is only possible using the RMT and not the transformer (§3.1).

We train a series of GPT2-small sized RMT models with residual stream sizes 384, 768, 1536, 2048, 3072, and 4096. All models are identical except for their residual key dimensions $D_k$. We choose to vary $D_k$ because it does not affect the model's core computations, allowing us to directly observe the effects of scaling the residual stream. The difference in total parameter and FLOP counts between the largest and smallest models is $< 1\%$.

The results are shown in Fig. 6. We see that increasing the residual stream size monotonically decreases dev loss, showing that expanding the residual stream improves model performance. If we compare the model with residual size 4096 to the model with residual size 768 (the residual stream size of GPT2-small), we see that the model with the expanded residual stream was able to train to the same final loss with 23% fewer FLOPS and 25% fewer tokens.

We can compare an RMT with a transformer with the same residual stream size, since the experiments performed in this section were trained with the same experimental settings as §4.2. We see that when the residual stream size is the same, the RMT model achieves a final train loss of 3.42 while the GPT2 model achieves 3.43. This verifies that if the residual stream is not expanded, the performance of the RMT is about the same as the transformer.

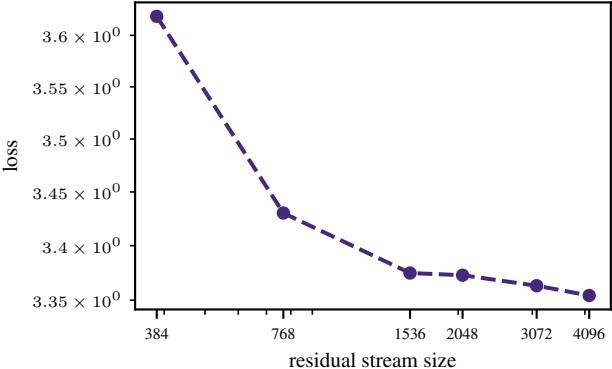

*Figure 6.* Dev loss vs residual stream size. Model size, dataset size, and computational budget are held constant for all runs. Residual stream size is calculated as $D_k \times D_v$.

### 4.5. Downstream Evaluation

In this section we present zero-shot downstream results of the RMT and transformer. We use our largest pretrained models from §4.2 trained on all 6B tokens. Table 5 and Table 6 show that the RMT performs significantly better

*Table 5.* Downstream perplexity results (Gokaslan et al., 2019, Paperno et al., 2016, Gao et al., 2020, Merity et al., 2016). Lower is better..

| DATA SET | TRANSFORMER | RMT |
|---|---|---|
| LAMBADA OPENAI | 61.34 | **25.21** |
| OPENWEBTEXT | 20.64 | **18.36** |
| PILE 10K | 30.23 | **23.54** |
| WIKITEXT | 76.35 | **55.94** |

*Table 6.* Downstream accuracy results (Clark et al., 2018, Zellers et al., 2019, Bisk et al., 2020, Sakaguchi et al., 2019). Higher is better.

| DATA SET | TRANSFORMER | RMT |
|---|---|---|
| ARC C | 19.7 | **20.6** |
| ARC E | 44.3 | **46.1** |
| HELLASWAG | 28.7 | **30.5** |
| PIQA | 61.9 | **63.7** |
| WINOGRANDE | 49.5 | **52.5** |

than the transformer on all downstream tasks despite having 25% fewer parameters and using 28% fewer FLOPS during training.

## 5. Related Work

Layer aggregation models bear some resemblance to our model, since they allow later layers to access earlier layers more easily. Indeed, our model can be interpreted as a type of layer aggregation model if one were to multiply out all key vectors and remove LayerNorm. Huang et al. (2017) introduced dense connections and showed that they can improve the performance of convolutional neural networks. Godin et al. (2017) extended dense connections to RNNs and showed that they can help with language modeling. Shen et al. (2018), Dou et al. (2018), and ElNokrashy et al. (2024) introduced dynamic layer aggregation through depthwise-attention and showed improvements on NMT and classification. Dou et al. (2018) also explored different types of static layer aggregation and showed improved results in NMT. Dou et al. (2019) then showed that layer aggregation with capsules and routing by agreement can further improve results on NMT. We note that for most layer aggregation models it is unrealistic to train in multi-gpu settings as they would incur a huge penalty in terms of device-to-device communication. Dou et al. (2018)'s model is an exception to this rule, and we include it in our comparison of transformer variants (§4.3).

The class of models that we find most similar to our own are those that manage their residual stream memory. These models offer multi-gpu scalability, and, in theory, they also make important features produced by earlier layers more

available to later layers. Srivastava et al. (2015) introduce Highway Networks that control which layer's outputs are carried down the residual stream. Chai et al. (2020) iterated on Highway Networks and applied them to transformers. Xu et al. (2024) used an LSTM to manage the residual stream. Our work is different from these as we consider the scaling ramifications of expanding the residual stream instead of managing its memory.

We find that modifications that involve residual stream scaling, normalization, and weight initialization to improve moment propagation (Kedia et al., 2024, Wang et al., 2024, Shleifer et al., 2021, Huang et al., 2020, Bachlechner et al., 2020) are orthogonal to our approach because they could be extended and applied to our model.

We consider transformer variants that modify components other than the residual stream (Dao & Gu, 2024, Peng et al., 2023, Sun et al., 2023, Fedus et al., 2022) to be orthogonal to our work as well since our approach can be easily integrated with these architectures.

## 6. Conclusion

We present a novel architecture that replaces the residual stream with an outer product memory mechanism, resulting in a transformer variant with an expandable residual stream. We theoretically show that our modifications provide efficient scaling and favorable moment propagation properties. Our architecture allows for scaling along a new scaling law axis that, to our knowledge, was previously unexplored. We experimentally showed that scaling the size of the residual stream leads to more efficient training in terms of compute, parameter, and data efficiency.

## Acknowledgments

We are thankful for the computing resources provided by the Pacific Research Platform's Nautilus cluster, supported in part by National Science Foundation (NSF) awards CNS-1730158, ACI-1540112, ACI-1541349, OAC-1826967, OAC-2112167, CNS-2100237, CNS-2120019, the University of California Office of the President, and the University of California San Diego's California Institute for Telecommunications and Information Technology/Qualcomm Institute, and CENIC for the 100Gbps networks. We also thank the anonymous ICML reviewers and the members of JLab for their proofreading and valuable feedback.

## Impact Statement

This paper presents work whose goal is to advance the field of Machine Learning. There are many potential societal consequences of our work, none which we feel must be specifically highlighted here.

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

# A. Parameter and FLOP calculations

See Table 7 for the equations used to compute FLOP and parameter counts for the transformer and RMT in Fig. 1. §A.1 and §A.2 describe how the equations for the RMT were derived. The equations for the transformer follow Hoffmann et al. (2024). Our derivations for the RMT's FLOP and parameter counts also follow the approach taken by Hoffmann et al. (2024). In particular, we approximate the backward-pass FLOP count as twice the forward-pass count.

## A.1. RMT Parameters

- Embeddings
    - Word embeddings: $RVD_v$
    - Position embeddings: $RND_v$
    - Key vectors: $2RD_k$
    - Embedding params: $RVD_v + RND_v + 2RD_k$

- Attention (Single Layer)
    - QKV key vectors: $3RD_k$
    - O key vectors: $RD_k$
    - Attention params: $3RD_k + RD_k$

- Feed-Forward Network (Single Layer)
    - Input key vectors: $RD_k$
    - Core params: $2RD_vD_{FF}$
    - Output key vectors: $RD_k$
    - FF params: $RD_k + 2RD_vD_{FF} + RD_k$

- Unembedding
    - Key vectors: $RD_k$
    - Unembedding weights: $RVD_v$
    - Unembedding params: $RD_k + RVD_v$

$$\text{Total Params} = \text{embedding params} + L \times \text{attention params} + \text{FF params}) + \text{unembedding params}$$
$$= R(2D_k(3L + 1) + D_v(2LD_{FF} + V + N))$$

## A.2. RMT FLOPS

- Embeddings
    - Word embeddings: $2NVRD_v$
    - Position embeddings: $2NVRD_v$
    - Key vectors: $4ND_kD_vR$
    - Embedding FLOPS: $2NVRD_v + 2NVRD_v + 4ND_kD_vR$

- Attention (Single Layer)
    - QKV key vectors: $6ND_kD_vR$
    - SHA: $2NND_vR + 3RNN + 2NND_vR$
    - O key vectors: $2ND_kD_vR$
    - Attention FLOPS: $6ND_kD_vR + 2NND_vR + 3RNN + 2NND_vR + 2ND_kD_vR$

- Feed-Forward Network (Single Layer)
    - Input key vectors: $2ND_kD_vR$

*Table 7.* Parameter and forward pass FLOP count equations.

| RESOURCE | TRANSFORMER | RMT |
|---|---|---|
| PARAMETERS | $D(L(4HD_h + 2D_{FF}) + V)$ | $R(2D_k(3L+1) + D_v(2LD_{FF} + V + N))$ |
| FLOPS | $4N(V + LD(2D_hH + D_{FF}) + L(ND_hH + \frac{3}{4}NH))$ | $6NRD_KD_v(1+L) + NR(4VD_v + L(4ND_v + 3N + 4D_vD_{FF}))$ |

- – Core params: $4NRD_vD_{FF}$

- – Output key vectors: $2ND_kD_vR$

- – FF FLOPS: $2ND_kD_vR + 4NRD_vD_{FF} + 2ND_kD_vR$

- Unembedding

  - – Key vectors: $2ND_kD_vR$

  - – Unembedding weights: $2NVRD_v$

  - – Unembedding FLOPS: $2ND_kD_vR + 2NVRD_v$

$$\text{Total forward pass FLOPS} = \text{embedding FLOPS} + L \times (\text{attention FLOPS} + \text{FF FLOPS}) + \text{unembedding FLOPS}$$
$$= 6NRD_KD_v(1+L) + NR(4VD_v + L(4ND_v + 3N + 4D_vD_{FF}))$$

## B. Moment Propagation Derivation

Here we modify the derivations from (Kedia et al., 2024) to derive the mean and variance propagation equations for the RMT's storage and retrieval operations.

### B.1. Storage

For $R$ input data vectors $\boldsymbol{x}_{\text{in}}^{(h)} \in \mathbb{R}^{D_v}$ and an output matrix $\boldsymbol{X}_{\text{out}} \in \mathbb{R}^{D_k \times D_v}$, the forward pass operation for a single token is defined as follows.

$$\boldsymbol{X}_{\text{out}} = \sum_{h=1}^{R} \boldsymbol{w}^{(h)} \otimes \boldsymbol{x}_{\text{in}}^{(h)}$$
$$= \sum_{h=1}^{R} \boldsymbol{w}^{(h)} \boldsymbol{x}_{\text{in}}^{(h)T}$$

The backward pass operation is then defined as follows.

$$\boldsymbol{g}_{\text{in}}^{(h)} = \boldsymbol{w}^{(h)T} \boldsymbol{G}_{out}$$

For the expected value of the forward pass, we have,

$$
\begin{aligned}
\mathbb{E}[\boldsymbol{X}_{\mathrm{out}_{ij}}] &= \mathbb{E}[\sum_{h=1}^{R} \boldsymbol{w}_i^{(h)} \boldsymbol{x}_{\mathrm{in}_j}^{(h)}] \\
&= \sum_{h=1}^{R} \mathbb{E}[\boldsymbol{w}_i^{(h)} \boldsymbol{x}_{\mathrm{in}_j}^{(h)}] \\
&= \sum_{h=1}^{R} \mathbb{E}[\boldsymbol{w}_i^{(h)}] \mathbb{E}[\boldsymbol{x}_{\mathrm{in}_j}^{(h)}] \quad \text{(by ind.)} \\
&= \sum_{h=1}^{R} \mathbb{E}[\boldsymbol{w}_i^{(h)}] \mathbb{E}[\boldsymbol{x}_{\mathrm{in}_j}^{(h)}] \\
&= \sum_{h=1}^{R} 0 \times \mathbb{E}[\boldsymbol{x}_{\mathrm{in}_j}^{(h)}] \quad \text{(w is initialized with mean 0)} \\
&= 0
\end{aligned}
$$

For the variance of the forward pass we have,

$$
\begin{aligned}
\mathrm{Var}(\boldsymbol{X}_{\mathrm{out}_{ij}}) &= \mathrm{Var}(\sum_{h=1}^{R} \boldsymbol{w}_i^{(h)} \boldsymbol{x}_{\mathrm{in}_j}^{(h)}) \\
&= \sum_{h=1}^{R} \mathrm{Var}(\boldsymbol{w}_i^{(h)} \boldsymbol{x}_{\mathrm{in}_j}^{(h)}) \quad \text{(by ind. of weights from each other)} \\
&= \sum_{h=1}^{R} ((\sigma_{x_{\mathrm{in}}}^2 + \mu_{x_{\mathrm{in}}}^2)(\sigma_w^2 + \mu_w^2) - \mu_{x_{\mathrm{in}}}^2 \mu_w^2) \quad \text{(by ind. of weights and inputs)} \\
&= \sum_{h=1}^{R} (\sigma_{x_{\mathrm{in}}}^2 + \mu_{x_{\mathrm{in}}}^2) \sigma_w^2 \\
&= R(\sigma_{x_{\mathrm{in}}}^2 + \mu_{x_{\mathrm{in}}}^2) \sigma_w^2
\end{aligned}
$$

For the expected value of the backward pass we have,

$$
\begin{aligned}
\mathbb{E}[\boldsymbol{g}_{\mathrm{in}_j}^{(h)}] &= \mathbb{E}[\sum_{i=1}^{D_k} \boldsymbol{w}_i^{(h)} \boldsymbol{G}_{\mathrm{out}_{ij}}] \\
&= \sum_{i=1}^{D_k} \mathbb{E}[\boldsymbol{w}_i^{(h)} \boldsymbol{G}_{\mathrm{out}_{ij}}] \\
&= \sum_{i=1}^{D_k} \mathbb{E}[\boldsymbol{w}_i^{(h)}] \mathbb{E}[\boldsymbol{G}_{\mathrm{out}_{ij}}] \quad \text{(by ind.)} \\
&= 0
\end{aligned}
$$

And for the variance of the backwards path we have,

$$\mathrm{Var}(\boldsymbol{g}_{\mathrm{in}_j}^{(h)}) = \mathrm{Var}(\sum_{i=1}^{D_k} \boldsymbol{w}_i^{(h)} \boldsymbol{G}_{\mathrm{out}_{ij}})$$

$$= \sum_{i=1}^{D_k} \mathrm{Var}(\boldsymbol{w}_i^{(h)} \boldsymbol{G}_{\mathrm{out}_{ij}}) \quad \text{(by ind. of weights from each other)}$$

$$= \sum_{i=1}^{D_k} ((\sigma_{G_{\mathrm{out}}}^2 + \mu_{G_{\mathrm{out}}}^2)(\sigma_w^2 + \mu_w^2) - \mu_{G_{\mathrm{out}}}^2 \mu_w^2) \quad \text{(by ind. of weights and inputs)}$$

$$= \sum_{i=1}^{D_k} (\sigma_{G_{\mathrm{out}}}^2 + \mu_{G_{\mathrm{out}}}^2) \sigma_w^2$$

$$= D_k (\sigma_{G_{\mathrm{out}}}^2 + \mu_{G_{\mathrm{out}}}^2) \sigma_w^2$$

## B.2. Retrieval

For $R$ input matrix $\boldsymbol{X}_{\mathrm{in}} = \in \mathbb{R}^{D_k \times D_v}$ and an output data vectors $\boldsymbol{x}_{\mathrm{out}}^{(h)} \in \mathbb{R}^{D_v}$, the forward pass operation for a single token is defined as follows.

$$\boldsymbol{x}_{\mathrm{out}}^{(h)} = \boldsymbol{w}^{(h)} \cdot_1 \boldsymbol{X}_{\mathrm{in}}$$
$$= \boldsymbol{w}^{(h)T} \boldsymbol{X}_{\mathrm{in}}$$

And for the backward pass we have,

$$\boldsymbol{G}_{\mathrm{in}} = \sum_{h=1}^{R} \boldsymbol{w}^{(h)} \boldsymbol{g}_{\mathrm{out}}^T$$

Notice that the forward equation for retrieval is the same as the backward table for storage, and vice versa. Then we can follow the exact same derivations as in §B.1 to get,

$$\mathbb{E}[\boldsymbol{x}_{\mathrm{out}}^{(h)}] = 0$$

$$\mathrm{Var}(\boldsymbol{x}_{\mathrm{out}}^{(h)}) = d_k \sigma_w^2 (\sigma_{X_{in}}^2 + \mu_{X_{in}}^2)$$

$$\mathbb{E}[\boldsymbol{G}_{\mathrm{in}}] = 0$$

$$\mathrm{Var}(\boldsymbol{G}_{\mathrm{in}}) = R \sigma_w^2 (\sigma_{g_{out}}^2 + \mu_{g_{out}}^2)$$

# C. Experimental Setup

## C.1. Training Details

All of our models are trained on the OpenWebText dataset and we use HuggingFace's GPT2 tokenizer. The vocab size is therefore set to 50257 for all models. Learned positional embeddings are used, and unless otherwise specified, models are trained with a sequence length of 512 tokens. We use pre-LayerNorm with the epsilon parameter set to $1e^{-6}$. We do not use any bias terms, nor do we use weight-tying for embedding and unembedding layers. In both models we include Mistral's GPT2 stability tweeks (attention upcasting and inverse layer scaling), as well as inverse layer scaling on layer outputs from GPT2. The loss function used is z-loss with its coefficient set to $1e^{-4}$. When we report loss values, we report them without the z-loss term which is equivalent to normal cross-entropy loss. All models are trained with the AdamW optimizer, with $\beta_1 = 0.9$, $\beta_2 = 0.95$, and $\epsilon = 1e^{-8}$. We use decoupled weight decay with a scaling factor of $1e^{-4}$. Layernorm,

*Table 8.* Transformer hyperparameters for scaling laws experiment

| MODEL SIZE | DEVICE | BATCH SIZE | $L$ | $D$ | $D_{FF}$ | $H$ | $D_h$ | TRAIN TOKENS |
|---|---|---|---|---|---|---|---|---|
| 49M | RTX 3090 | 64 | 6 | 384 | 1536 | 12 | 32 | 212M |
| 160M | RTX 3090 | 64 | 12 | 768 | 3072 | 12 | 64 | 1.7B |
| 260M | A100 | 64 | 18 | 896 | 3584 | 14 | 64 | 3.5B |
| 405M | A100 | 256 | 24 | 1024 | 4096 | 16 | 64 | 6B |

*Table 9.* RMT hyperparameters for scaling laws experiment

| MODEL SIZE | DEVICE | BATCH SIZE | $L$ | $D_k$ | $D_v$ | $R$ | $D_{FF}$ | TRAIN TOKENS |
|---|---|---|---|---|---|---|---|---|
| 46M | RTX 3090 | 64 | 6 | 32 | 32 | 12 | 1536 | 142M |
| 134M | RTX 3090 | 64 | 12 | 32 | 64 | 12 | 3072 | 1.1B |
| 206M | A100 | 64 | 18 | 896 | 48 | 14 | 3584 | 2.3B |
| 305M | A100 | 256 | 24 | 1024 | 64 | 16 | 4096 | 6B |

embedding, and unembedding weights are excluded when weight decay is applied. We use linear warmup for $5\%$ of the total training tokens, and cosine decay for the remaining tokens that decays to $10\%$ of the max learning rate. In addition, gradient checkpoint is used with all models.

All training experiments are implemented using JAX (Bradbury et al., 2018), Equionx (Kidger & Garcia, 2021), and Haliax (Hall et al., 2024).

## C.2. RMT vs Transformer Experiments

Table 8 and Table 9 show the hyperparameters used for the experiments in §4.2.

### C.2.1. $\mu$PARAM

The "Parameterization Lottery" refers to how new architectures can appear to be successful or not successful based on their compatibility with existing hyperparameters. Furthermore, the typical hyperparameters used when training standard transformers are the product of years of tuning done by the machine learning community. To help mitigate biases associated with hyperparameter selection, we perform our own hyperparameter tuning on both the transformer and RMT's using $\mu$Transfer. For each model type we perform 100 hyperparameter searches that follow suggestions from the $\mu$Transfer paper. We sample from a loguniform distribution: learning rate on the interval $[1e^{-4}, 1e^{-1}]$, input alpha on the interval $[1e^{-1}, 1.]$, attention alpha on the interval $[1e^{-1}, 1.]$, output alpha on the interval $[1e^{-1}, 1.]$, and initialization standard deviation on the interval $[1e^{-1}, 1.]$. Our trail models have sequence length 128, 4 layers, 8 attention heads, head dimension 32, and 1024 feed-forward neurons. The embed size of the standard transformer is set to 256, and for the RMT the residual key and value dimensions are both set to 32, and the layer rank $R$ is set to 8. These models are all trained on Nvida GTX 1080 Ti gpus with a batch size of 32, train over 5K steps. We select the hyperparameters of the best performing run from each model type. For the standard transformer we have lr $= 6.5e^{-3}$, input alpha $= 9.76$, attention alpha $= 0.25$, output alpha $= 5.52$, and init std $= 0.15$.

## C.3. Scaling Residual Stream Experiments

These models are all trained on Nvidia GeForce RTX 3090 gpus with a batch size of 64. All models use a max learning rate of $1e^{-2}$ and are trained over 50K steps. The models have 12 layers, residual matrix value dimension $D_v$ of 64, 3072 feed-forward neurons, and a layer rank $R$[8] of 12 for a total of 135M parameters. Note that $R$ is equal to the number of attention heads, $R * D_v$ is input/output dimension of the core feed-forward operation (Eq. 24), and $D_v$ is the attention head dimension.

---

[8]Same $R$ from Eqs. in §2.2

*Table 10.* Transformer variants statistics

| ARCHITECTURE | MODEL SIZE | RESIDUAL STREAM SIZE | FINAL TRAIN LOSS |
|---|---|---|---|
| TRANSFORMER | 160M | 768 | 3.43 |
| RMT | 134M | 2048 | 3.38 |
| HIERARCHICAL AGGREGATION | 165M | 768 | 3.47 |
| DEPTHWISE LSTM | 233M | 768 | 3.54 |
| HIGHWAY TRANSFORMER | 191M | 768 | 3.41 |

### C.4. Transformer Variants Experiments

All models were trained on Geforce-RTX 3090 gpus with a batch size a 64. They were trained for 5000 train steps for a total of 1.6B train tokens. The models all had 12 layers, $D_{FF}$ of 3072, 12 attention heads, and $D_h$ = 64. For the RMT model $D_k$ was set to 32 and $D_v$ was set to 64. Table 10 shows the models size and residual stream sizes for these models. The max learning rate was set to $1e^{-2}$ and we used Lecun initialization for all parameters.

## D. Downstream Evaluation Details

All results from Table 6 plus the Lambada perplexities presented in Table 5 were generated using the EleutherAI evaluation harness (Gao et al., 2023). The remaining perplexity results in Table 5 were evaluated by hand.

## E. Runtime Discussion

Although hardware-specific implementations and diagnostics are outside the scope of this work, we hypothesize that much of the slowdown is due to replacing highly optimized GEMM operations with unoptimized tensor contractions. The storage and retrieval operations are both implemented as tensor contractions, whereas the storage and retrieval operations for the transformer are GEMMs. The contraction dimension of the RMT's tensor contraction is small compared to the embedding dimension of the transformer (the K dimension of the transformer's GEMM operations). We expect that a naive implementation of this tensor contraction will have much lower throughput than an optimized GEMM kernel; however, we expect that a custom CUDA kernel that takes advantage of the small size of the key vectors could dramatically improve runtime performance.

