# OpenReview forum: "Residual Matrix Transformers: Scaling the Size of the Residual Stream"
_ICML.cc/2025/Conference — ICML 2025 poster_

### Official Review · Reviewer_ciWT · 2025-03-11

**Overall Recommendation:** 3

**Summary:**

The Residual Matrix Transformer (RMT) replaces the residual stream in transformers with an outer product memory matrix, allowing independent scaling of the residual stream size. This results in improved training efficiency and better performance on downstream tasks.

**Claims And Evidence:**

Yes

**Essential References Not Discussed:**

None

**Experimental Designs Or Analyses:**

Yes

**Methods And Evaluation Criteria:**

Yes

**Other Comments Or Suggestions:**

None

**Other Strengths And Weaknesses:**

1. Runtime efficiency concerns: The authors acknowledge that runtime is currently the biggest limitation of their model, with RMT being 4% slower than the transformer despite being more FLOP-efficient. This suggests potential implementation inefficiencies that could limit practical adoption.
2. Limited model size exploration: Due to resource constraints, the authors couldn't explore how efficiency trends continue at larger model sizes beyond 405M parameters, leaving questions about scalability to truly large models.
3. Baseline Comparisons: The transformer variants in §4.3 (Dou et al., 2018; Xu et al., 2024) are not state-of-the-art (e.g., recent works like Mamba or RWKV). Including stronger baselines would better contextualize RMT’s advancements.

**Questions For Authors:**

None

**Relation To Broader Scientific Literature:**

None

**Theoretical Claims:**

None

---

> ### Author Rebuttal · Authors · 2025-03-27
>
> We thank Reviewer ciWT for their comments and provide responses to their concerns about runtime efficiency and baseline comparisons.
> For concerns about runtime, we will copy and paste a relevant snippet of our response to reviewer J7dP. We encourage Reviewer ciWT to read our discussion with Reviewer J7dP if more context is needed.
>
>
> We do not think runtime is an insurmountable obstacle for our model for two reasons. The first is that, without any hardware optimizations, the RMT only takes 4% more time to achieve the same train loss as the Transformer. This shows that the performance gain needed is within reasonable reach. The second reason is that, while the residual stream size is much larger in the RMT, many of the parameter matrices are much smaller. One can use this fact to devise more efficient kernels for the RMT. For example, many of the “key” parameters are so small that they can fit entirely into SMEM. One can then imagine a GEMM kernel that only has to reload one operand matrix into SMEM, significantly decreasing the data transfer overhead.
>
> We further assert that the main focus of our paper is to explore residual stream size as a new scaling axis. While we do discuss runtime in the interest of full transparency, we consider the discussion of specific hardware optimizations to be out of the scope of this work.
>
> Next, we address concerns about baseline comparisons.
> > The transformer variants in §4.3 (Dou et al., 2018; Xu et al., 2024) are not state-of-the-art (e.g., recent works like Mamba or RWKV). Including stronger baselines would better contextualize RMT’s advancements.
>
> We consider works like Mamba and RWKV to be orthogonal to our work because they primarily change the attention layer of the Transformer. Because these architectures use the residual stream in the same way the Transformer does, our method can be extended to be integrated with them. We agree with the reviewer that some clarifying discussion should be added to our paper.

---

### Official Review · Reviewer_J7dP · 2025-03-12

**Overall Recommendation:** 3

**Summary:**

The paper introduces Residual Matrix Transformers (RMT), which increases the size of the residual stream in a transformer without incurring significant compute over memory overhead by using an outer product memory matrix. In training GPT-2 language models, RMT achieves better loss per unit of compute or parameters.

**Claims And Evidence:**

The GPT-2 experiments support the claim that RMT outperforms the standard transformer, but experiments on other datasets (e.g. images) would be helpful for assessing the generalizability of the finding.

**Essential References Not Discussed:**

None that I'm aware of.

**Experimental Designs Or Analyses:**

The use of µP for estimating optimal learning rates for different models is not necessarily sound, given that the training iterations are not constant across model sizes, which is an assumption in µP. In addition, I'm concerned that the larger learning rate used for RMT may unfairly favor it over standard transformers. Overall I think a more careful learning rate sweep is necessary to demonstrate the superiority of RMT convincingly.

**Methods And Evaluation Criteria:**

The GPT-2 experiment on OpenWebText is a reasonable benchmark. But it would be useful to understand if applying RMT still improves performance when applied on top of the modern transformer architecture actually used in practice such as the one used in llama models (with rotary embedding, SwiGLU activation, RMSNorm instead of LayerNorm and no biases).

**Other Comments Or Suggestions:**

While the authors do not address the problem of the slower runtime of RMT, I believe it is an important question and it is unclear whether the runtime overhead of RMT can be addressed even in principle due to the additional data transfer. I suggest that the authors analyze and discuss whether the runtime overhead of RMT can, in fact, be addressed and by what kind of approaches. For example, if most of the overhead is in the additional data transfer involving the larger residual stream matrix, that overhead may in fact be irreducible without changing the hardware.

**Other Strengths And Weaknesses:**

Strength: Scaling the residual stream size is a novel idea and appears to be a promising research direction.

Weakness: RMT makes multiple modifications to the transformer architecture. These modifications do not represent the unique way of scaling the residual stream size and deserve to be motivated better or be ablated to illustrate which components actually matter for its performance.

**Questions For Authors:**

1. When using µP for estimating optimal learning rates, do you properly scale the found learning rate per layer from smaller to larger models?

**Relation To Broader Scientific Literature:**

Finding new axes worth scaling beyond the usual ones, like compute and parameters, is an important research direction. The finding that scaling the residual stream size alone leads to considerable performance gains is interesting and relevant to the community. On the other hand, as the authors discussed, this approach increases the memory overhead and data transfer, which can translate to slower runtimes. While RMT performs better than standard transformers when controlling for FLOPs, ultimately, we care about performance per unit of time, and using fewer FLOPs is not relevant if the hardware utilization is compromised. Indeed, operations like attention are often memory-bound rather than compute-bound, and techniques that reduce the data transfer can significantly speed up the runtimes even while increasing the FLOPs, as exemplified by FlashAttention [1].

[1] Dao, Tri, et al. "Flashattention: Fast and memory-efficient exact attention with io-awareness." Advances in neural information processing systems 35 (2022): 16344-16359.

**Theoretical Claims:**

N/A

---

> ### Author Rebuttal · Authors · 2025-03-27
>
> We thank reviewer J7dP for their thoughtful reading of the paper and appreciate their comment that this work presents an important research direction. We provide responses to many of their concerns grouped by subject.
>
> We first address the concerns related to our application of µP transfer.
>
> > The use of µP for estimating optimal learning rates for different models is not necessarily sound, given that the training iterations are not constant across model sizes, which is an assumption in µP.
>
> We argue that using µP to transfer hyperparameters across training iterations is reasonable given the empirical evidence provided in the µP paper. The authors empirically showed that their method works across sequence length, depth, batch size, and training time (see Table 1 of their paper). In fact, the only hyperparameter that the paper theoretically proved transferability across was model width. In their GPT-3 experiment (the closest setting to ours), the authors successfully transfered hyperparameters from a proxy model trained with significantly fewer training iterations. Given the success of their experiment, we felt that it was reasonable to take the same approach that they did.
>
> > In addition, I'm concerned that the larger learning rate used for RMT may unfairly favor it over standard transformers. Overall I think a more careful learning rate sweep is necessary to demonstrate the superiority of RMT convincingly.
>
> In our main comparison between the RMT and the Transformer (§4.2), we included learning rate as one of the hyperparameters in our μP search and used the best performing learning rates of both the RMT and transformer models.
>
> > When using µP for estimating optimal learning rates, do you properly scale the found learning rate per layer from smaller to larger models?
>
> We scale learning rate in accordance with Table 8 and Appendix B.1 of the µP paper (i.e. by $\frac{1}{\textrm{width ratio}}$ for hidden parameters).
>
>
> Here we give a response to the reviewer’s concerns about the runtime of our model.
>
> We do not think runtime is an insurmountable obstacle for our model for two reasons. The first is that, without any hardware optimizations, the RMT only takes 4% more time to achieve the same train loss as the transformer. This shows that the performance gain needed is within reasonable reach. The second reason is that, while the residual stream size is much larger in the RMT, many of the parameter matrices are much smaller. One can use this fact to devise more efficient kernels for the RMT. For example, many of the “key” parameters are so small that they can fit entirely into SMEM. One can then imagine a GEMM kernel that only has to reload one operand matrix into SMEM, significantly decreasing the data transfer overhead.
>
> We further assert that the main focus of our paper is to explore residual stream size as a new scaling axis. While we do discuss runtime in the interest of full transparency, we consider the discussion of specific hardware optimizations to be out of the scope of this work.
>
> Finally, we respond to the following concern:
> > RMT makes multiple modifications to the transformer architecture. These modifications do not represent the unique way of scaling the residual stream size and deserve to be motivated better or be ablated to illustrate which components actually matter for its performance.
>
> We acknowledge the concern that RMT introduces several modifications to the transformer architecture that may seem arbitrary without proper justification. However, we would emphasize that these changes represent the minimal adjustments necessary to support the outer product memory matrix structure of its residual stream. Each modification is integral to the model's coherent functioning - removing any single element would create architectural inconsistencies. While we certainly don't claim RMT represents the only possible approach to scaling residual stream size, our work demonstrates one viable method for doing so, allowing us to investigate the effects of scaling along this axis.

---

### Official Review · Reviewer_fZLC · 2025-03-15

**Overall Recommendation:** 3

**Summary:**

To achieve more data-efficient and compute-efficient models, this paper introduces a new transformer-variant called the Residual Matrix Transformer (RMT), which replaces the traditional residual stream with an outer product memory matrix. The authors present theory showing that the RMT exhibits efficient scaling of the residual stream and improved variance propagation properties in some cases.
Experimental results demonstrate that when using an RMT with a larger residual stream size, it is more efficient than traditional transformer models in terms of data, FLOPS, and parameters. Additionally, it proves to be superior in performance compared to various transformer variants. Ultimately, the paper shows that increasing the residual stream size leads to better model performance.

**Claims And Evidence:**

Most of the claims made in the submission are supported by clear and compelling evidence, but some of them also need further justification (see Experimental Designs Or Analyses).

**Essential References Not Discussed:**

Yes, there are essential references that could enhance understanding.

**Experimental Designs Or Analyses:**

- The experiments may lack some ablation studies to demonstrate the contributions of different factors. For instance, the authors need to demonstrate whether the superior performance of RMT over the Transformer stems from the outer product memory matrix structure or is simply a result of the increased residual stream size. In the main comparison, RMT outperforms the Transformer, but its residual stream is 2.5 to 4 times larger, which could also be a contributing factor to the observed performance gains. Beyond the analysis provided in §3.1 of the paper, thorough ablation experiments are crucial to validate these claims.
- The paper states in lines 81-83 that "RMT has improved variance propagation." However, Table 2 reveals that RMT underperforms the Transformer in the Attention retrieval case. These "bad cases" need further analysis to provide readers with a comprehensive understanding of the proposed method's performance and its potential application scenarios.

**Methods And Evaluation Criteria:**

The proposed methods and evaluation criteria are well-suited for the problem and application at hand.

**Other Comments Or Suggestions:**

No

**Other Strengths And Weaknesses:**

No

**Questions For Authors:**

No

**Relation To Broader Scientific Literature:**

The key contributions of the paper align well with existing literature, building on prior findings and enhancing our understanding of the topic.

**Theoretical Claims:**

I checked the proofs for the theoretical claims, and they were correct and well-justified.

---

> ### Author Rebuttal · Authors · 2025-03-27
>
> We appreciate the reviewer fZLC’s thoughtful feedback and positive comments. We will address each of this reviewer’s concerns in the order they appear.
>
> > The experiments may lack some ablation studies to demonstrate the contributions of different factors. For instance, the authors need to demonstrate whether the superior performance of RMT over the Transformer stems from the outer product memory matrix structure or is simply a result of the increased residual stream size.
>
> This is a valid point that we will make more clear in the paper.  In our experiments, we find that the performance gains are not attributable to the outer product memory matrix structure itself, but rather to the expanded residual stream that this structure enables. One of the RMT models in §4.4 has the same residual stream size as the Transformer in §4.3, and these models are trained with the exact same experimental settings. The final train loss that the RMT achieves is 3.42 while the final train loss that the Transformer achieves is 3.43, showing that when the residual stream size is the same, the observed performance is about the same. These results suggest that the observed performance gains of the RMT over the Transformer in §4.1 are due to the expanded residual stream.
>
> > In the main comparison, RMT outperforms the Transformer, but its residual stream is 2.5 to 4 times larger, which could also be a contributing factor to the observed performance gains. Beyond the analysis provided in §3.1 of the paper, thorough ablation experiments are crucial to validate these claims.
>
> As mentioned above, the observed performance gains are most likely due to the residual stream being 2.5 to 4 times larger. We would like to clarify, however, that the ability to expand the residual stream to this size is made possible by the RMT’s outer product memory matrix structure. Expanding the residual stream of the Transformer would substantially increase the model’s parameter count and per-example compute cost. As such, an ablation where the Transformer’s residual stream size is expanded but the parameter count, FLOP count, and tokens consumed is fixed is not possible.
>
> > The paper states in lines 81-83 that "RMT has improved variance propagation." However, Table 2 reveals that RMT underperforms the Transformer in the Attention retrieval case. These "bad cases" need further analysis to provide readers with a comprehensive understanding of the proposed method's performance and its potential application scenarios.
>
> This is also a valid point, however we consider that performing a full analysis of the end-to-end signal propagation through the RMT to be out of the scope of this paper (for example, an entire paper was dedicated to this for the Transformer (Kedia et al., 2024)). We maintain that improving the signal propagation of 3 out of the 4 replaced components roughly shows that our model has superior signal propagation properties compared to the Transformer. We also want to note that we found an error in the submitted manuscript. In Table 2, the attention storage and retrieval numbers are swapped, i.e. the RMT outperforms the Transformer in the Attention retrieval case and underperforms the Transformer in the Attention storage case. An correction to the table is provided below.
>
>
> \begin{array} {|r|r|}\hline Layer & Operation & Model & \frac{\sigma^{2}\_{x\_{out}}}{\sigma^{2}\_{x\_{in}}} & \frac{\sigma^{2}\_{g\_{in}}}{\sigma^{2}\_{g\_{out}}} \\\\ \hline Attn & Storage & RMT & 0.4 & 1.6 \\\\ \hline Attn & Storage & Transformer & 1 & 1 \\\\ \hline Attn & Retrieval & RMT & 1.14 & 0.86 \\\\ \hline Attn & Retrieval & Transformer & 0.5 & 1.5 \\\\ \hline  \end{array}
>
> > Yes, there are essential references that could enhance understanding.
>
> We have included all the references that we find enhance the understanding of our work, and we welcome any specific suggestions for additional sources that would strengthen the manuscript.

---

### Decision · Program_Chairs · 2025-05-01

**Decision:**

Accept (poster)

**Comment:**

All reviewers argued for acceptance but only one engaged after the rebuttal period (J7dp). They had lingering concerns about whether hyperparameters introduced an bias that would favour RMT and whether the overhead of RMT could be significantly reduced with a better implementation.